# Irrigation Intensities Drive Soil N_2_O Emission Reduction in Drip-Irrigated Cotton Fields

**DOI:** 10.3390/plants14070987

**Published:** 2025-03-21

**Authors:** Honghong Ma, Qi Wu, Xianglin Wu, Qianqian Zhu, Shenghai Pu, Xinwang Ma

**Affiliations:** 1Institute of Agricultural Resources and Environment, Xinjiang Academy of Agricultural Sciences, Urumqi 830091, China; mahh@xaas.ac.cn (H.M.); wuxianglin@xaas.ac.cn (X.W.); zhuqq2022@xaas.ac.cn (Q.Z.); 2Key Laboratory of Northwest Oasis Agriculture Environment, Ministry of Agriculture, Urumqi 830091, China; 3College of Land Science and Technology, China Agricultural University, Beijing 100193, China; wu1120170375@163.com

**Keywords:** cotton field, drip irrigation, irrigation intensity, microorganisms, influencing factors, N_2_O emission

## Abstract

Drip irrigation with plastic mulch is widely used to save water and improve fertilizer efficiency in arid regions in Xinjiang. However, farmers freely use irrigation water in pursuit of a high cotton yield, and the impact of different irrigation amounts on nitrous oxide (N_2_O) emissions is still unclear. A field experiment was conducted in 2023 in Xinjiang, China, with drip-irrigated cotton (*Gossypium hirsutum* L.) to determine N_2_O emissions with different irrigation intensities. The different irrigation treatments were designed as follows: irrigation was performed to maintain soil moisture at (1) an 80% field capacity (Q80); (2) 90% field capacity (Q90); and (3) 100% field capacity (Q100). The results showed that the yield of cotton decreased with the increase in irrigation intensity. A 100% field capacity is beneficial for ammonium and nitrate transformation. The N_2_O emissions remained at a relatively low level during the non-irrigated fertilization period. In every irrigation and fertilization cycle, the N_2_O emissions were mainly concentrated during the process from wet to dry. The peak occurred during days 1–3 of irrigation. Throughout the growth period, the cumulative N_2_O emissions were 1.15, 1.48, and 2.63 kg N ha^−1^ under the Q80, Q90, and Q100 treatments, respectively. As the irrigation intensity increased, the dominant species of soil bacteria and fungi showed substitution, while the dominant species of soil actinomycetes were not replaced. Fungi, actinomycetes, the available potassium, and the carbon to nitrogen ratio were positively correlated with nitrous oxide emissions, and the soil temperature was negatively correlated with nitrous oxide emissions. These results demonstrate that increased irrigation could increase the risk of greenhouse gas emissions when using plastic mulch with drip irrigation.

## 1. Introduction

Nitrous oxide (N_2_O) is a potent greenhouse gas (GHG) that has existed for almost 120 years in the atmosphere, which also depletes stratospheric ozone and drives global climate change [1,2,3]. Nitrogen (N) fertilizer input is the main source of the N_2_O emissions of agricultural soils, accounting for 60% of the total global anthropogenic flux [4,5,6]. N_2_O is produced through biotic and abiotic processes, including ammonia oxidation, byproducts of ammonia oxidation, autotrophic denitrification, chemical denitrification, and heterotrophic denitrification [1,2,7,8]. The application of chemical nitrogen fertilizers stimulates these processes, and nitrous oxide emissions increase with the increase in nitrogen fertilizer applications [3,9]. In addition, these processes are regulated by climatic factors and soil properties, such as the type and amount of nitrogen fertilizer, soil organic carbon (SOC), soil moisture, soil temperature, and soil pH [1,2,10].

Cotton (*Gossypium hirsutum* L.) is the main cash crop in Xinjiang, Northwestern China. According to the data provided by the National Bureau of Statistics of China, the planting area in 2022 was 2369.3 thousand ha in Xinjiang, accounting for 84.98% of the national cotton planting area [11]. Due to the arid climate and limited water resources, under-membrane drip irrigation technology has been widely adopted in Xinjiang, which can transport fertilizers together with water into the root of the crop so as to achieve a uniform, accurate, timed, and rationed supply of water and nutrients, reduce water evaporation, and increase soil temperature [12]. Nitrogen fertilizer (usually urea) is dissolved with drip irrigation and applied in batches along with irrigation water through surface or subsurface drip irrigation systems for multiple applications during the growing season. Thus, the soil undergoes a process of drying to wetting and then wetting to drying after each irrigation and fertilizer application. High-frequency wet–dry alternation led to higher N_2_O emissions due to enhanced microbial activity and increased nitrogen mineralization rates [4]. Wet–dry alternation may increase, decrease, or have no effect on soil N_2_O emissions, with studies reaching inconsistent conclusions [4].

It is hypothesized that a lower irrigation intensity can reduce nitrous oxide emissions, which directly reduces the contribution of agricultural production activities to global warming and helps to alleviate the severe water shortage in Xinjiang, Northern China. This study aims to explore the characteristics of nitrous oxide emission under different irrigation intensities and identify the main factors driving nitrous oxide emissions in drip-irrigated cotton fields in Xinjiang. The results will provide a theoretical basis for optimizing irrigation management practices to reduce soil N_2_O emissions, enhance nitrogen use efficiency, and support sustainable agricultural production under the double-carbon strategy.

## 2. Materials and Methods

### 2.1. Site Description and Soil Properties

Experiments were conducted in 2023 in cotton field in Baotou Lake farm in Korla City, Xinjiang, China (41.69° N, 85.87° E) (Figure 1). The average annual rainfall is 56.2 mm, and the average annual evaporation is 2497.4 mm. The average number of annual sunshine hours is 2878 h. The effective accumulated temperature is 4252.2 °C. The meteorological data during the planting period are presented in Figure 2 (LI-7500DS, Lincoln, NE, USA). The frost-free period is 205 days. The groundwater level is from 2.0 to 2.5 m. The soil texture is sandy loam soil, with a medium fertility level. Field soil belongs to Typic Haploxeralfs [13]. The bulk density and field capacity of the 0–30 cm soil are 1.33 g·cm^−3^ and 23.77%. The properties of the surface soil (0–30 cm) at the experimental site were as follows: organic matter is 8.10 g·kg^−1^, total N is 0.8 g·kg^−1^, available nitrogen (AN) is 89.24 mg·kg^−1^, available phosphorus (AP) is 71.45 mg·kg^−1^, available potassium (AK) is 272.67 mg·kg^−1^, and pH level is 8.20.

### 2.2. Experimental Design and Agronomic Management

The field experiment was conducted from April to October 2023. Three irrigation treatments were designed: irrigation was performed to maintain soil moisture at (1) 80% field capacity (Q80); (2) 90% field capacity (Q90); and (3) 100% field capacity (Q100). The selection of these irrigation intensities was based on common practices in the region, with 90% field capacity representing the typical irrigation intensity used by local farmers to maximize cotton yield. Irrigation time was every 7 days, which aligns with the daily water and fertilizer management practices of local farmers. Irrigation was applied 11 times throughout the cotton’s entire growth period. The first drip irrigation began on June 19th and the last irrigation ended on August 28th. The specific irrigation times and amounts are shown in Table 1.

Each treatment had 360 kg N ha^−1^ urea (46% N), 225 kg P ha^−1^ calcium phosphate (44% P), and 150 kg K ha^−1^ K_2_SO_4_ (50% K), which were based on the local cotton fertilizer consumption. Granular urea was applied where 20% of the total N was spread onto the field by hand as basal fertilizer before sowing for the three treatments. The remaining 80% of solubilized N was applied in the second to 11th irrigation events as follows: 4%, 6%, 6%, 10%, 24%, 10%, 6%, 6%, 4%, and 4%, respectively. P as calcium phosphate and K as K_2_SO_4_ were spread by hand onto all the plots, and they were placed into the soil using a rotator before sowing. The treatments were laid out according to a randomized complete block design with three replications. Each plot was 4.5 m × 8.0 m.

Cotton cultivar Xinluzzhong 56, a high-yielding variety developed for Xinjiang with high level of environmental adaptability, high lint percentage, disease/pest resistance, and superior fiber quality, was sown in late April. Drip irrigation pipes and plastic mulch were used during the sowing process, which was implemented using a custom-built tractor-drawn seeder. Seeds were sown in double rows with a gap of 30 cm between the two rows that formed a pair and a gap of 60 cm between one pair and the next. Within each row, seeds were sown 10 cm apart. The plastic mulch is made up of high-density, airtight, transparent polythene film in strips wide enough to cover two double rows. Weeds, pests, and disease were controlled using commercial herbicides, insecticides, and pesticides for local management of cotton. The cotton bud period was from July 3rd to August 3rd, flowering and boll period was from August 4th to September 4th, and boll-opening period was from September 5th to October 5th.

### 2.3. Sampling and Analysis

We collected soil samples from the 0–30 cm soil layer on the day before and three days after each irrigation application throughout the cotton growing season. The sampling locations are directly below the drip head, 38 cm to the left of the drip irrigation belt, and 38 cm to the right of the drip irrigation belt. After being thoroughly mixed, the soil is taken and divided into two parts. A portion of the soil sample is used to determine soil moisture content. The other portion is placed in a refrigerator for measuring soil nitrate nitrogen and ammonium nitrogen. We took a small amount of a soil sample before the experiment began and a final soil sample to determine the physical and chemical properties, which were analyzed at the Key Laboratory of Northwest Oasis Agriculture Environment, Ministry of Agriculture (Urumqi, China). We harvested fresh soil samples of 0–30 cm to determine soil microbial indicators.

We weighed fresh soil samples to obtain 0.5 g of each, passed them through a sieve (<1 mm), and removed plant debris and soil fauna. Total DNA was extracted to determine soil bacterial (16S rRNA genes), fungal (ITS regions), actinomycete (16S rRNA genes), ammonia-oxidizing archaea (AOA, 16S rRNA genes), and ammonia-oxidizing bacteria (AOB, 16S rRNA genes) contents using high-throughput sequencing methods (Illumina) and quantitative PCR (qPCR) techniques [14,15,16].

Soil bulk density (BD) is determined by ring knife method (100 cm^3^) [17]. Soil field capacity is determined by drying method [18]. Soil nitrate and ammonium nitrogen are determined by continuous flow injection analyzer (Seal AA3, Norderstedt, Germany). Soil temperature and moisture are determined by online monitoring system of three soil parameters (Campbell CS655, Logan, UT, USA). Soil organic matter (SOM) is determined using the potassium dichromate wet-combustion method [19]. Soil pH is determined using the potentiometric method. Soil total carbon (C) and nitrogen (N) are determined by elemental analyzer (Elementar Inc., Hanau, Hessen, Germany). Soil total phosphorus (P) is determined by visible spectrophotometer (Santa Clara, CA, USA). Soil available phosphorus (AP) is determined by the Olsen method [20]. Soil available potassium (AK) is determined by the ammonium acetate extraction method, followed by flame photometry to measure the extracted potassium [21]. Soil available potassium (K) is determined by potassium acetate extraction method. N_2_O flux in soil–air interface is determined by enhanced desktop N_2_O analyzer (GLA351, Pittsburgh, PA, USA).

All cotton plants were manually harvested at the end of the growing season, and cotton bolls were collected from each plant in each experimental plot. Bolls of diameter >2 cm were categorized as mature bolls, <2 cm as young bolls, and shell cracks >3 mm as flocculent bolls. Rotten bolls were excluded from statistics. We put the cotton bolls in the same experimental plot together and weighed them (electronic scale, 0.1 kg). Subsequently, the bolls were processed using a 4MZ-6 cotton gin (China) to physically separate lint cotton from cottonseeds.

The yield per hectare (t/ha) was calculated using the following formula:$$Y=\frac{m}{A}\times 10$$

*Y* = *Y* represents yield per unit area (t ha^−1^)*m* represents total weight of seed or lint cotton in the test plot (kg), and *A* represents total area of the test plot (m^2^).

### 2.4. Data Analysis

Raw data were organized using Excel software (Microsoft Corporation, Redmond, WA, USA). Ggplot2 package in R software (Version 4.3.2, R Foundation for Statistical Computing, Vienna, Austria) was used to draw line and bar charts to test significant differences in cumulative N_2_O emissions among different treatments, using a one-way analysis of variance (ANOVA). Before performing ANOVA, the homogeneity of variances among the groups was assessed using Levene’s test. When the assumption of homogeneity of variance was met (*p* > 0.05), ANOVA F-test for intergroup comparisons was used; otherwise, one-way ANOVA was employed for the analysis. To determine the factors affecting cumulative N_2_O emissions, the correlation between cumulative N_2_O emissions and the tested indicators were calculated using the Pearson correlation function cor in R, followed by visualization with ggplot2. Furthermore, linear regression analysis was performed between N_2_O flux and soil volumetric water content. All statistical results are presented as the mean value ± standard error, with significance levels set at *p* < 0.05 and *p* < 0.001 to indicate statistical differences.

## 3. Results

### 3.1. Cotton Yield

There were significant differences in the seed cotton yield and lint yield under different irrigation intensities, with their overall performances ranked as follows: Q90 > Q80 > Q100 (Table 2). The yield of unginned and ginned cotton under Q90 increased by 39.6% and 42.0%, respectively, compared with Q80. The yield of unginned and ginned cotton under Q100 decreased by 36.1% and 34.0%, respectively, compared with Q80. In general, the yield of cotton decreased with the increase in irrigation intensity.

### 3.2. $NH_{4}^{+}$-N and $NO_{3}^{-}$
^_^N

The content of $NH_{4}^{+}$-N was closer to 2.50 mg·kg^−1^ before irrigation under the three treatments, with the values of 2.50 (Q80), 2.47 (Q90), and 2.52 mg·kg^−1^ (Q100), respectively (Figure 3A,C,E). After irrigation, the contents of $NH_{4}^{+}$-N had the following order: Q100 > Q90 > Q80, with the values of 6.00, 5.72, and 5.45 mg·kg^−1^, respectively. There was a significant increase in $NH_{4}^{+}$-N before and after the irrigation. The ranges of $NH_{4}^{+}$ for Q80, Q90, and Q100 during the entire irrigation period were 2.06–5.45 mg·kg^−1^, 2.16–5.72 mg·kg^−1^, and 2.27–6.00 mg·kg^−1^. The result indicated that a 100% field capacity is beneficial for ammonium nitrogen transformation.

The content of $NO_{3}^{-}$-N was closer to 22.53 mg·kg^−1^ before irrigation under the three treatments, with the values of 22.90 (Q80), 22.54 (Q90), and 22.14 mg·kg^−1^ (Q100), respectively (Figure 3 B,D,F). After irrigation, the contents of $NO_{3}^{-}$-N had the following order: Q100 > Q90 > Q80, with the values of 124.60, 119.18, and 108.34 mg·kg^−1^, respectively. There was a significant increase in $NO_{3}^{-}$-N before and after the irrigation. The variation ranges of $NO_{3}^{-}$-N during the entire irrigation period were 22.90–108.34 mg·kg^−1^ (Q80), 22.54–119.18 mg·kg^−1^ (Q90), and 22.14–124.60 mg·kg^−1^ (Q100). The result indicated that a 100% field capacity is beneficial for nitrate nitrogen transformation.

### 3.3. N_2_O Flux

The N_2_O emissions remained at a relatively low level during the non-irrigated fertilization period, ranging from 0.02 to 0.79, 0.01 to 0.50, and 0.02 to 0.73 mg·m^−2^ under Q80, Q90, and Q100, respectively (Figure 4). The N_2_O emissions were mainly concentrated in the irrigation and fertilization period from July to August. In every irrigation and fertilization cycle, the N_2_O emissions were mainly concentrated in the wet–dry period. The peak occurred during 1–3 days after irrigation, with the values of 22.80 mg·m^−2^ (Q80), 32.94 mg·m^−2^ (Q90), and 62.98 mg·m^−2^ (Q100), respectively. In general, the N_2_O emissions showed an increase first and then a decreasing trend during the entire irrigation and fertilization period, which was consistent with the trend of the irrigation and fertilization amounts. As the irrigation amount increased, the N_2_O emissions also gradually increased, indicating that irrigation intensity is an important factor affecting N_2_O emissions. Throughout the cotton growth period, the cumulative N_2_O emissions were 1.15, 1.48, and 2.63 kg N ha^−1^ under Q80, Q90, and Q100, respectively.

The accumulated N_2_O emissions during the bud stage were the highest, followed by the flowering and boll stage, and finally the opening stage (Figure 5). The cumulative emissions of N_2_O in the three growth stages under Q80 accounted for 73.18%, 23.84%, and 2.98% of total emissions, respectively. The cumulative emissions of N_2_O in the three growth stages under Q90 accounted for 78.32%, 16.88%, and 4.80% of total emissions, respectively. The cumulative emissions of N_2_O in the three growth stages under Q100 accounted for 65.18%, 31.94%, and 2.88% of total emissions, respectively. As a whole, there was a significant difference across the entire growth period under different irrigation intensities. With the increase in irrigation intensity, the cumulative emissions of N_2_O increase.

### 3.4. Microbial Community Under Different Irrigation Treatments

#### 3.4.1. Bacterium

There were 108 bacterial species in the Q80 treatment, 122 in the Q90 treatment, and 136 in the Q100 treatment (Figure 6). A total of 42 bacterial species were found in the Q80, Q90, and Q100 treatments. A total of 34 endemic species in Q80, 48 endemic species in Q90, and 56 endemic species in Q100 were found. The top two bacteria with the highest relative abundance were *Rhodovulum* sp. and *Arenimonas* sp. under the Q80 treatment, *Rhodovulum* sp. and *Lysobacter fragaria* under the Q90 treatment, and *Arenimonas* sp. and *bacterium E2-14* under the Q100 treatment, respectively. As the irrigation intensity increased, the number of bacterial species and individual species increased. The relative abundance of *bacterium E2-14* and *Nocardioides gangwensis* increased with the increase in irrigation intensity, while the relative abundance of *Rhodovulum* sp. decreased slightly. With the increase in irrigation, the dominant species of soil bacteria changed, indicating that the irrigation intensity had a significant effect on the soil bacterial community’s structure.

#### 3.4.2. Fungi

The number of soil fungal species was 370 in the Q80 treatment, 273 in the Q90 treatment, and 536 in the Q100 treatment (Figure 7). A total of 124 fungal species were found in the Q80, Q90, and Q100 treatments. A total of 122 species were unique to Q80, 63 to Q90, and 270 to Q100. The two fungi with the highest relative abundances were *Pseudoeurotium desertorum* and *Penicillium rubidurum* under the Q80 treatment and the Q90 treatment. The two fungi with the highest relative abundances were *Fusarium equiseti* and *Hebeloma mesophaeum* under the Q100 treatment. The number of fungal species and individual species decreased and then increased with increasing irrigation intensity. The relative abundances of *Pseudoeurotium desertorum* and *Penicillium rubidurum* gradually decreased, while the relative abundance of *Fusarium equiseti* gradually increased to become the dominant species under the Q100 treatment. With the increase in irrigation, the dominant species of soil fungi changed, indicating that the irrigation intensity had a significant effect on the soil fungal community’s structure.

#### 3.4.3. Actinobacteria

There were seven species of actinomycetes in Q80, seven species of actinomycetes in Q90, and thirteen species of actinomycetes in Q100 (Figure 8). Three fungal species were common to Q80, Q90, and Q100, two were endemic to Q80, one was endemic to Q90, and nine were endemic to Q100. The highest relative abundance of actinomycetes under the three treatments was found in *Rhacophorus dennysi*, while the relative abundance of the other types of actinomycetes was relatively low. As the irrigation intensity increased, the number of actinomycete species gradually increased, and the dominant species under the different treatments did not change, indicating that the irrigation intensity has a relatively small impact on the structure of actinomycete communities.

### 3.5. The Main Factors

After performing the Pearson correlation analysis, we found significant correlations (|r| > 0.8) between cumulative nitrous oxide (N_2_O) emissions and several environmental and biological factors. The results of the analysis are as follows:

Positive and strong correlations: The cumulative N_2_O emissions showed very strong positive correlations with the fast-acting potassium content (K, r = 0.95), ammonium nitrogen ($NH_{4}^{+}$,-N r = 0.83), the carbon/nitrogen ratio (C/N, r = 0.93), and microbial abundance (bacteria (BAC), fungi (ITS), and actinomycetes (ACT); r = 0.83–1.00). These results suggest that N_2_O production and release may be synergistically promoted in environments high in organic matter and rich in ammonium nitrogen, as well as in the presence of active microbial activity.

Negative strong correlation: Significant negative correlations were exhibited between the cumulative N_2_O emissions and the soil temperature (Temp, r = −0.98) and the total nitrogen content (N, r = −0.80). This suggests that N_2_O production and emissions may be suppressed under high-temperature conditions, thereby reducing gaseous losses of nitrogen (Figure 9).

## 4. Discussion

### 4.1. The Effects of Irrigation Intensities on N_2_O Emissions

Nitrous oxide production, consumption, and transmission are directly or indirectly influenced by irrigation regimes. In our study, the N_2_O emissions were mainly concentrated on the period of irrigation and fertilization from July to September, especially from July to August. In each irrigation and fertilization cycle, the N_2_O emissions were mainly concentrated during the wet–dry period after irrigation and fertilization. The peak of N_2_O emissions occurs between the day of irrigation and the third day after irrigation. Multiple lines of evidence suggest that alternating wet and dry irrigation increases nitrogen mineralization rates and N_2_O emissions [22]. Wet–dry alternation affects N_2_O emissions by influencing the soil aggregate structure, void size, and microbial communities, and its effect depends mainly on the frequency and intensity of wet–dry alternation. The highest production of N_2_O in soil occurred in soil water-filled pore spaces (WFPSs): 50–70% [23]. However, the highest production of N_2_O occurred under a 100% field capacity in our study, which may be because the increase in WFPS in soil limits the utilization of oxygen and provides more favorable conditions for denitrification or nitrifying bacteria, thereby increasing soil N_2_O emissions [24]. When the soil volume moisture content exceeds a 60% field capacity, N_2_O mainly comes from denitrification. This result is inconsistent with Sha et al. [25]. Sha et al. [25] found that a lower soil water content promoted N_2_O production. The increase in N_2_O flux after basal fertilizer application was mainly attributed to denitrification. And the discovery by Johannes et al. [26] also illustrates this point. In the process of denitrification, microbial mortality was increased, and organic matter was decomposed due to frequent wet–dry alternation [27]. Drip irrigation systems have specific requirements for soil micro-environmental conditions such as soil moisture, temperature, microbial communities, and activity [28].

The mean N_2_O emissions in cotton under drip irrigation with plastic mulch were 1.75 kg N ha^−1^ in the growing season, and the N_2_O emissions increased with increasing irrigation amounts (Figure 3). The emissions were much lower than those in croplands, with an average of 2.55 kg N ha^−1^ in North America and Europe, and 2.10 kg N ha^−1^ was recorded in the fertilized soils of cotton fields according to 192 N_2_O area datasets [29,30]. Ma et al. [30] found growing season N_2_O emissions ranged between 259 and 473 g N ha^−1^ with 240 kg N ha^−1^ urea and an irrigation amount of 450 mm with plastic mulch and drip fertigation in Xinjiang, which were lower than our result, probably because of the following: (1) the amount of nitrogen fertilizer is two-thirds of that in our study; and (2) N_2_O emissions increase exponentially with nitrogen application rate according to Shcherbak et al. [31]. Thus, the amount of nitrous oxide emissions cannot easily be quantified, which is closely related to water and fertilizer management measures, soil texture, nutrient content, and other factors.

### 4.2. The Main Factors Influencing N_2_O Emissions

Previous studies have shown that there is a negative relationship between the soil C/N ratio and N_2_O emissions, mainly because a higher soil C/N ratio favors complete denitrification; i.e., N_2_O is reduced to N_2_ [12,24]. But the soil carbon and nitrogen ratios had a positive relationship with N_2_O emissions in our study, mainly because soil carbon and nitrogen ratios affect the composition and activity of the microbial community, which in turn affects the production of N_2_O, which indicates an increase in the soil nitrogen retention capacity and a reduction in nitrogen losses [32]. Some evidence suggests that fungi are powerful nitrifying and denitrifying organisms in arid and semi-arid soils, while archaea also play a significant role in nitrification and denitrification processes [33,34,35,36]. Fungi play a major role in N_2_O production, followed by archaea and bacteria [36]. This finding is in agreement with our results. Some fungi have a complete denitrification pathway that produces an intermediate product, N_2_O, during denitrification. Actinomycetes can carry out nitrification to convert ammonium nitrogen to nitrate nitrogen, and N_2_O is produced in this process. The soil water content affects the oxygen status of the soil, and it is easier to form an anaerobic environment when the water is saturated, which is conducive to the occurrence of the denitrification process. Although an elevated temperature can promote microbial activity and accelerate the rate of nitrogen conversion, it also promotes the further reduction of N_2_O to N_2_, thus showing a negative correlation.

### 4.3. Limitations and Future Works

This study did not monitor post-basal-fertilization soil N_2_O emission fluxes, which should be prioritized in subsequent experiments to fully assess fertilizer-driven greenhouse effects. Additionally, while microbial community structure was analyzed post-irrigation, long-term dynamic monitoring across growth stages is lacking; future work should incorporate multi-temporal soil sampling and high-throughput sequencing to track microbial evolution. Laboratory simulations are also needed to identify key mechanisms linking irrigation/fertilization practices to N_2_O production, coupled with physiological and biochemical assays to elucidate microbial metabolic pathways.

## 5. Conclusions

The yield of cotton decreased with the increase in irrigation intensity. Specifically, maintaining the soil moisture at a 100% field capacity (Q100) resulted in the lowest cotton yield, while a 90% field capacity (Q90) achieved the highest yield. This indicates that excessive irrigation can negatively impact cotton productivity. Additionally, a 100% field water holding capacity was found to be beneficial for ammonium nitrogen and nitrate nitrogen transformation in drip-irrigated cotton fields in Northwest China, Xinjiang. This suggests that while a higher irrigation intensity can enhance nitrogen transformation processes, it may not always be optimal for crop yield.

The N_2_O emissions remained at a relatively low level during the non-irrigated fertilization period. In every irrigation and fertilization cycle, the N_2_O emissions were mainly concentrated during wet–dry period, with peaks occurring 1–3 days after irrigation. The cumulative N_2_O emissions increased with the increase in irrigation intensity, reaching 1.15, 1.48, and 2.63 kg N ha^−1^ under the Q80, Q90, and Q100 treatments, respectively. As the irrigation intensity increased, the dominant species of soil bacteria and fungi changed, while the dominant species of soil actinomycetes did not. Fungi, actinomycetes, the available potassium, and the carbon to nitrogen ratio were positively correlated with nitrous oxide emissions, while the soil temperature was negatively correlated with nitrous oxide emissions. These findings demonstrate that increased irrigation can increase the risk of greenhouse gas emissions when using plastic mulch with drip irrigation. Therefore, optimizing irrigation intensity is crucial for reducing N_2_O emissions and mitigating the environmental impact of agricultural practices.

## Figures and Tables

**Figure 1 plants-14-00987-f001:**
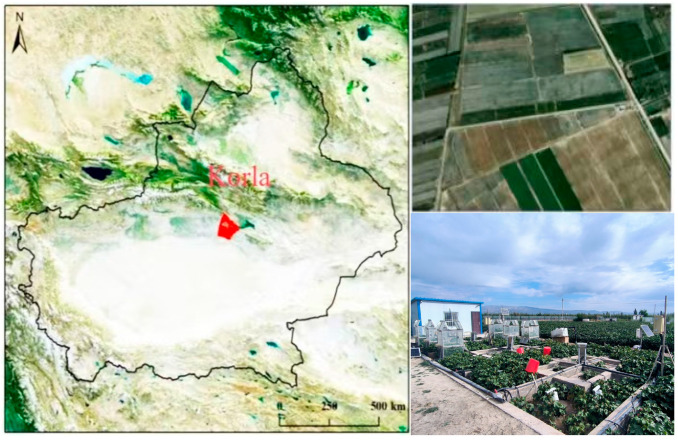
Experimental sites.

**Figure 2 plants-14-00987-f002:**
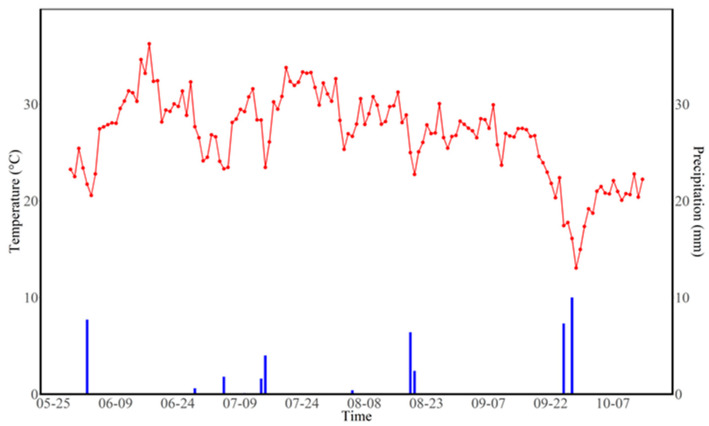
Meteorological data during the planting period.

**Figure 3 plants-14-00987-f003:**
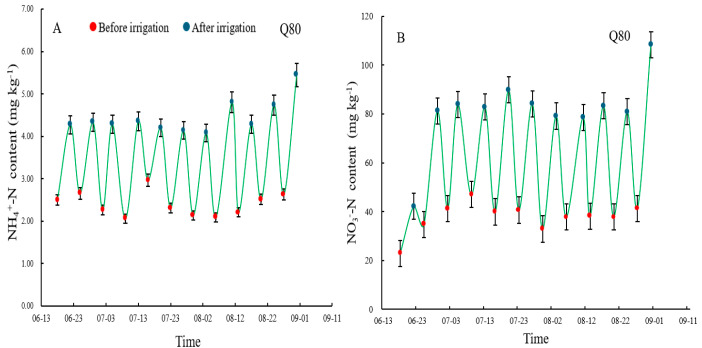
Dynamics of soil ammonium nitrogen and nitrate nitrogen before and after irrigation. (**A**,**B**) show the content of ammonium nitrogen and nitrate nitrogen under Q80 treatment; (**C**,**D**) under Q90 treatment; (**E**,**F**) under Q100 treatment. Red dots represents pre-irrigation. Blue dots represents post-irrigation.

**Figure 4 plants-14-00987-f004:**
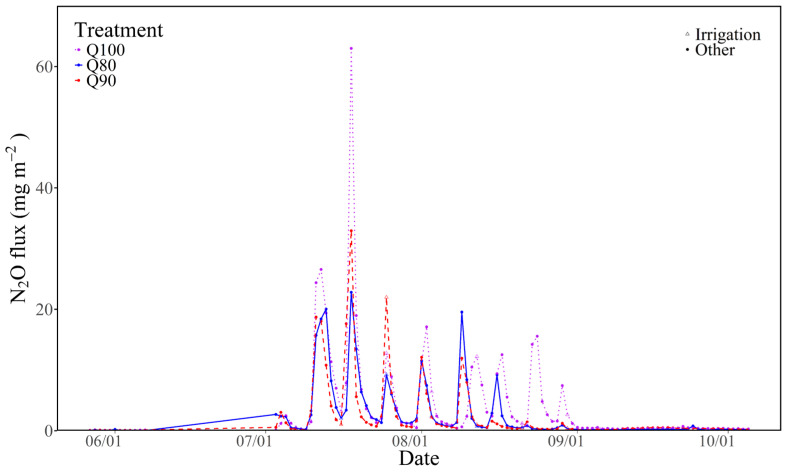
Emissions of nitrous oxide (N_2_O) during the irrigation period in cotton field under different irrigation intensities.

**Figure 5 plants-14-00987-f005:**
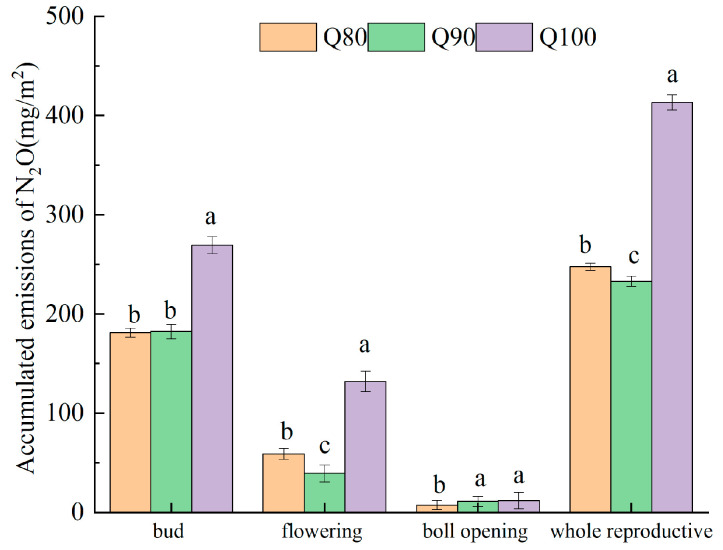
Accumulated emissions of N_2_O in cotton field under three irrigation intensities. Different lowercase letters indicate differences between different treatments at a *p*-value of 0.05.

**Figure 6 plants-14-00987-f006:**
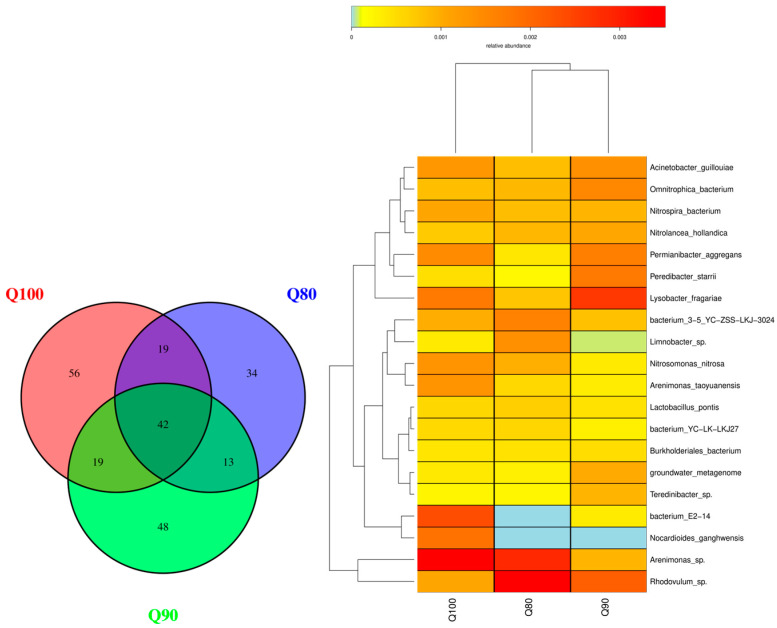
Bacterial distribution under three irrigation intensities in cotton field.

**Figure 7 plants-14-00987-f007:**
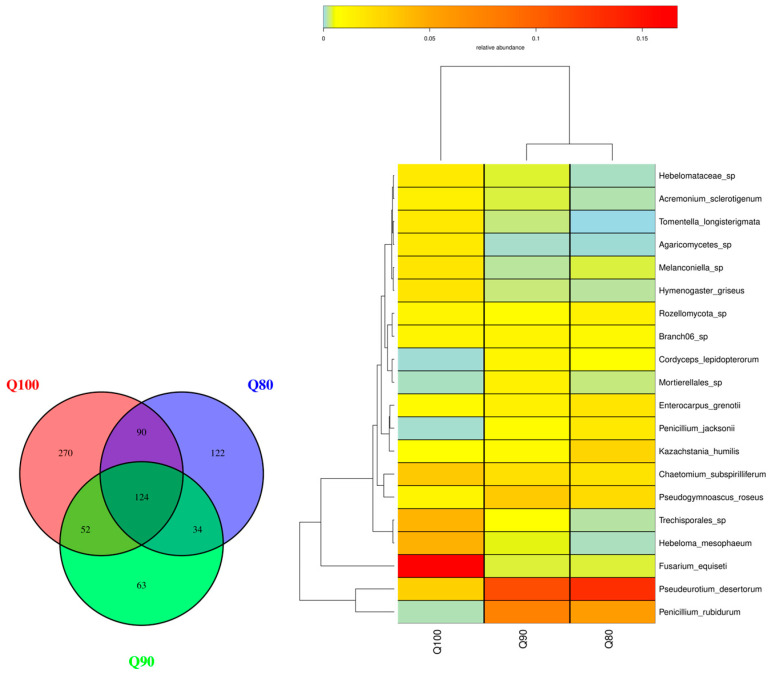
Fungal distribution under three irrigation intensities.

**Figure 8 plants-14-00987-f008:**
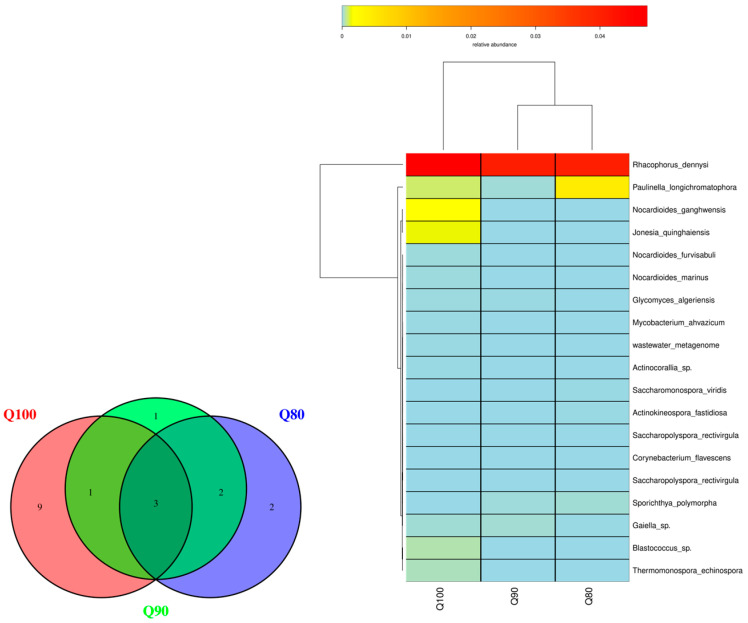
Actinobacteria’s distribution under three irrigation intensities.

**Figure 9 plants-14-00987-f009:**
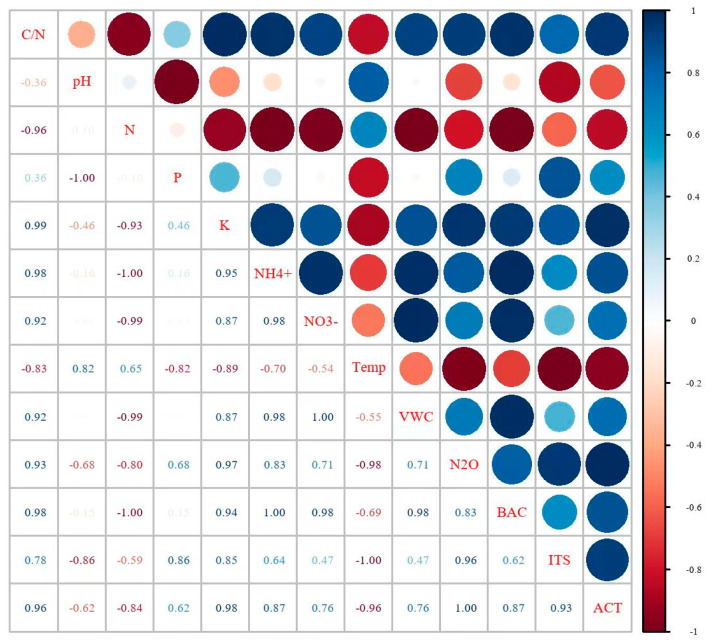
Correlation coefficient between cumulative N_2_O emissions and measured indicators under different irrigation intensities. $NH_{4}^{+}$-N represents the ammonium nitrogen content, $NO_{3}^{-}$-N represents the nitrate nitrogen content, Soil Tc represents soil temperature, Soil VWC represents soil volumetric moisture content, BAC represents bacteria, ITS represents fungi, and ACT represents actinomycetes.

**Table 1 plants-14-00987-t001:** Irrigation amount applied per treatment (mm).

Treatment	1st	2nd	3rd	4th	5th	6th	7th	8th	9th	10th	11th	Total Amount
Date	19 June	26 June	3 July	10 July	17 July	24 July	31 July	7 August	14 August	21 August	28 August	
Q80	7	11	18	44	47	47	58	55	36	22	11	358
Q90	9	14	24	57	62	62	76	71	48	29	14	466
Q100	12	18	30	71	77	77	95	89	59	36	18	580

**Table 2 plants-14-00987-t002:** Seed cotton yield and lint cotton yield under different treatments (t ha^−1^).

Treatment	Unginned Cotton	Ginned Cotton
Q80	1.242 ± 0.019 b	1.734 ± 0.027 b
Q90	1.763 ± 0.018 a	2.421 ± 0.025 a
Q100	0.793 ± 0.014 c	1.144 ± 0.020 c
	*p* value	
Intensity	0.03	<0.01
	*F* value	
Intensity	10.16	36.65

Note: Different lowercase letters indicate significant differences between treatments (*p* < 0.05).

## Data Availability

Data will be made available on request.

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
