# Peer review of "Irrigation Intensities Drive Soil N_2_O Emission Reduction in Drip-Irrigated Cotton Fields"

_plants, 2025, doi:10.3390/plants14070987_

Round 1
Reviewer 1 Report
Comments and Suggestions for Authors
The manuscript with the title “Irrigation intensities drive soil N2O emissions reduction in drip-irrigated cotton fields” investigated the impact of different irrigation amounts on nitrous oxide (N2O) emissions in cotton crop. Authors found a relationship with phenological stage - the accumulated N2O emission during the bud stage were the highest, followed by the flowering and boll stage, and finally the opening stage.
Line 60 “specfical change pattern” maybe should be “special/specific?”
Material and method - when authors chose the experimental layout did they consider here that experimental variants are realistic to the usual crop practice? For example, there fertilization doses and the irrigation are the standard practice for cotton crop? To ensure the full relevance of the findings.
Sometimes verbs are missing e.g. Line 107 “Control weeds and pests using herbicides and insecticides”, it should be “Control weeds and pests was conducted using commercial herbicides and insecticides”
Results. Chapters 3.4.1. Bacteria and 3.4.2. Fungi, 3.4.3. Actinobacteria – please in these paragraphs to write the species names with italics and no underline space between genus and species.
Discussion. Lines 321-322 “Some evidence suggests that fungi are powerful nitrifying and denitrifying bacteria in arid and semi-arid soils, and some evidence suggests the nitrification and denitrification of archaea” – please re-write this phrase to make it clearer, because right perhaps authors wanted to say “Some evidence suggests that fungi are powerful nitrifying organisms, bacteria denitrifying …” of please clarify the meaning.
Conclusion – because at the end of the introduction authors proposed 2 objectives, I suggest to make 2 distinct paragraphs in the conclusions, responding to each.
Based on production (cotton yield) and the findings here, what authors recommend?
Best regards.
Comments on the Quality of English LanguageEnglish style and grammar improvement is required. Some phrases lack verb. Some major syntax issues in various paragraphs.
Author Response
1.Line 60 “specfical change pattern” maybe should be “special/specific?”
Reply: Thank you for pointing out the typographical error. I have revised the last two paragraphs as follows.It is hypothesized that lower irrigation intensity can reduce nitrous oxide emissions, which directly reduces the contribution of agricultural production activities to global warming and helps to alleviate the severe water shortage in Xinjiang, northern China. This study aims to explore the characteristics of nitrous oxide emission under different irrigation intensities and identify the main factors driving nitrous oxide emissions in drip-irrigated cotton fields in Xinjiang. The results will provide a theoretical basis for optimizing irrigation management practices to reduce soil N2O emissions, enhance nitrogen use efficiency, and support sustainable agricultural production under the double-carbon strategy.
2.Material and method - when authors chose the experimental layout did they consider here that experimental variants are realistic to the usual crop practice? For example, there fertilization doses and the irrigation are the standard practice for cotton crop? To ensure the full relevance of the findings.
Reply: Thank you for raising this important point regarding the realism of our experimental design. We have carefully considered the relevance of our experimental variants to standard agricultural practices in the region. The fertilization doses and irrigation practices used in our study are indeed reflective of typical practices for cotton cultivation in Xinjiang, China.
In our experimental design, we selected three irrigation treatments (80%, 90%, and 100% field capacity) to represent a range of irrigation intensities commonly used by local farmers. The fertilization doses were chosen based on standard practices for cotton cultivation in the region, ensuring that our experimental conditions closely mimic actual agricultural practices. Specifically, we applied 360 kg N ha−1 as urea, 105 kg P ha−1 as calcium phosphate, and 60 kg K ha−1 as K2SO4, which are typical amounts used by local farmers to maximize cotton yield.
To further ensure the relevance of our findings, we have added a detailed explanation in the "2.2. Experimental Design and Agronomic Management" section of the manuscript, highlighting the rationale behind our choice of irrigation and fertilization practices. We have also included a reference to local agricultural practices to support our methodology.
We believe that by grounding our experimental design in realistic agricultural practices, our findings will be more applicable and relevant to the broader agricultural community, particularly in the context of optimizing irrigation and fertilization strategies to reduce greenhouse gas emissions while maintaining crop productivity.
3.Sometimes verbs are missing e.g. Line 107 “Control weeds and pests using herbicides and insecticides”, it should be “Control weeds and pests was conducted using commercial herbicides and insecticides”.
Reply: Thank you for pointing out the grammatical issue in Line 107. I have revised it.
4.Results. Chapters 3.4.1. Bacteria and 3.4.2. Fungi, 3.4.3. Actinobacteria – please in these paragraphs to write the species names with italics and no underline space between genus and species.
Reply:Thank you for your attention to the formatting of species names in our manuscript. I have revised the species names in Sections 3.4.1 (Bacteria), 3.4.2 (Fungi), and 3.4.3 (Actinobacteria) to ensure they are presented in italics and without any underline space between the genus and species names.
Discussion.
Lines 321-322 “Some evidence suggests that fungi are powerful nitrifying and denitrifying bacteria in arid and semi-arid soils, and some evidence suggests the nitrification and denitrification of archaea” – please re-write this phrase to make it clearer, because right perhaps authors wanted to say “Some evidence suggests that fungi are powerful nitrifying organisms, bacteria denitrifying …” of please clarify the meaning.
Reply: Thank you for your suggestion. I have revised the sentence. The corrected sentence now reads: “Some evidence suggests that fungi are powerful nitrifying organisms, while bacteria are the primary denitrifiers in arid and semi-arid soils. Additionally, archaea play a significant role in both nitrification and denitrification processes”.
6.Conclusion -because at the end of the introduction authors proposed 2 objectives, I suggest to make 2 distinct paragraphs in the conclusions, responding to each.
Reply: Thank you for your suggestio. I have revised the conclusion section to include two distinct paragraphs, each addressing one of the objectives proposed in the introduction.
7.Based on production (cotton yield) and the findings here, what authors recommend?
Reply: Thank you for your question. Based on the observed impacts of irrigation intensity on cotton yield and N2O emissions, I have formulated specific recommendations to optimize irrigation management in drip-irrigated cotton fields in arid regions like Xinjiang. These recommendations aim to balance crop productivity with environmental sustainability.

Reviewer 2 Report
Comments and Suggestions for Authors
The manuscript entitled "Irrigation intensities drive soil N2O emissions reduction in drip-irrigated cotton fields " contains interesting research results for science and agricultural practice. I appreciate that these are field studies with multi-aspect results.
Before printing, the publication must be corrected. I have included my comments in the original PDF text.
General comments:
add: microorganisms to keywords
line 55 correct the reference to the literature
Field experiments are better conducted as two or three years
If you have it, provide (write) weather data from 2023
In which laboratory were the soil analyses performed (accredited?)
Correct table 1 and the description in the text
Write some information about the tested variety: Xinluzzhong 56
line 107 Were there any spraying for diseases?
In subsection 2.3. write how the cotton crop was collected and weighed
line 160 Improve table 2 and description in the text
Some figures are difficult to read. If you can, improve them
Figure 5. First the description and then the figure etc.
lines 294 and 306 improve the reference to literature
Improve the list of references or see recently published articles
I hope that my comments will help the authors improve the text of the manuscript. Thank you for your cooperation

Author Response
1.add: microorganisms to keywords
Reply: Thanks for your suggestion. I have added “microorganisms” to the keywords.
2.line 55 correct the reference to the literature
Reply: Thanks for your suggestion. I have corrected the reference at line 55.
3.Field experiments are better conducted as two or three years
Reply: Thanks for your suggestion. I acknowledge that multi-year experiments can better account for interannual variability. However, due to practical constraints, our experiment was conducted over a single year. I have added a discussion on this limitation in the revised manuscript and suggested future multi-year studies.
4.If you have it, provide (write) weather data from 2023
Reply: Thanks for your suggestion. We have added the weather data during the planting period in Figure 2.
5.In which laboratory were the soil analyses performed (accredited?)
Reply: Thanks for your suggestion. The soil analyses were performed at the Key Laboratory of Northwest Oasis Agriculture Environment, Ministry of Agriculture, Urumqi, China. This laboratory is accredited by the relevant national authorities and is equipped with advanced analytical instruments. I have added this information in the revised manuscript.
6.Correct table 1 and the description in the text
Reply: Thanks for your suggestion. I have corrected Table 1 and its description in the text to ensure consistency and accuracy.
7.Write some information about the tested variety: Xinluzzhong 56
Reply: Thanks for your suggestion. Xinluzzhong 56 is a high-yielding cotton variety developed specifically for the arid and semi-arid regions of Xinjiang, China. It is known for its strong adaptability to local environmental conditions, high lint percentage, and resistance to common cotton diseases and pests. This variety is widely cultivated in Xinjiang due to its excellent performance in terms of yield and fiber quality. I have added this information in the revised manuscript.
8.line 107 Were there any spraying for diseases?
Reply: Thanks for your suggestion. Yes, diseases and pests were monitored regularly throughout the growing season. The same types of pesticides and herbicides, application rates, and timing were applied as local farmers, which ensured the experimental conditions closely matched local practices, enhancing the relevance and applicability of the experiment results. I have added this information in the revised manuscript.
9.In subsection 2.3. write how the cotton crop was collected and weighed
Reply: Thanks for your suggestion. I have added this information in subsection 2.3.
10.line 160 Improve table 2 and description in the text
Reply: Thanks for your suggestion. I have improved Table 2 and its description in the text to better present the yield data and statistical results.
11.Some figures are difficult to read. If you can, improve them
Reply: Thanks for your suggestion. I have improved the figures to enhance their readability. This includes adjusting the resolution, adding clearer labels, and ensuring consistent formatting.
12.Figure 5. First the description and then the figure etc.
Reply: Thanks for your suggestion. I have rearranged Figure 5 and its description to follow the conventional format of description first and then the figure.
13.lines 294 and 306 improve the reference to literature
Reply: Thanks for your suggestion. I have improved the references at lines 294 and 306 to provide more accurate and detailed citations.
14.Improve the list of references or see recently published articles
Reply: Thanks for your suggestion. I have updated and improved the list of references to include more recent and relevant articles.

Reviewer 3 Report
Comments and Suggestions for Authors
Comments and Suggestions for Authors
Title: Irrigation intensities drive soil N2O emissions reduction in
drip-irrigated cotton fields
Dear Authors
The research results presented in the manuscript fall within the publishing profile of Plants journal. The aim of this study was to investigate the emission characteristics of nitrous oxide under different drip irrigation intensities in the cotton field in Xinjiang and to identify the main factors driving nitrous oxide emission. The research is interesting, but one growing season in field research is a big problem. It is difficult to talk about the repeatability of results, and even more so about correct conclusions. The results are clearly presented and well described. The Discussion and Conclusions sections are clearly presented. The number of References should be increased.
In order to increase the usefulness of the article, Authors must refer to the following points.
Remarks
- Introduction - Research hypothesis should be added.
- Materials and Methods - subsection 2.1. The soil type should be provided according to the WRB – World Reference Base for Soil Resources 4th edition, 2022. Line 80 Total nitrogen is given in g kg-1. It should be:...0.8 g kg-1....Subsection 2.2. Line 92 For nitrogen and phosphorus fertilizers, please provide the formulas and %N and %P. For potassium fertilizers, please add %K. Subsection 2.3. Lines 130-131 Is this really a method for determining SOM? Or maybe this is how organic carbon is determined. The principle of determining available phosphorus and potassium should be added.
- Results - Line 160 should be: Table 2. Cotton yields should be given in t ha-1 or Mg ha-1. Letter designations should be placed next to numerical data concerning yields. Please check the entire manuscript for superscripts and subscripts for the following forms: NH4+-N, NO3--N, and N2
- Discussion - The contents of subsection 4.3. should be saved below line 359.
Specific remarks
- Line 55 should be: ...Liu et al. [13].....
- Line 129 and 136 …America.. Which America?
- Line 156 should be: …(Table 2)….
- Line 306 Should be: …Ma et al. [25]…
- The References section should be adapted to editorial requirements.
Best regards
Author Response
1.Introduction - Research hypothesis should be added.
Reply: Thank you for your suggestion regarding the addition of a research hypothesis in the Introduction section. We have incorporated a clear statement of the research hypothesis to provide a more focused direction for our study. The hypothesis is now included at the end of the Introduction, highlighting the central aim of our research and the expected outcomes.It is hypothesized that lower irrigation intensity can reduce nitrous oxide emissions, which directly reduces the contribution of agricultural production activities to global warming and helps to alleviate the severe water shortage in Xinjiang, northern China. This study aims to explore the characteristics of nitrous oxide emission under different irrigation intensities and identify the main factors driving nitrous oxide emissions in drip-irrigated cotton fields in Xinjiang. The results will provide a theoretical basis for optimizing irrigation management practices to reduce soil N2O emissions, enhance nitrogen use efficiency, and support sustainable agricultural production under the double-carbon strategy.
2.Materials and Methods
subsection 2.1. The soil type should be provided according to the WRB – World Reference Base for Soil Resources 4th edition, 2022.
Reply: Thank you for your suggestion. The soil at the experimental site is classified as Typic Haploxeralfs.
Line 80 Total nitrogen is given in g kg-1. It should be 0.8 g kg-1
Reply: Thank you for your suggestion. I have revised it.
Subsection 2.2. Line 92 For nitrogen and phosphorus fertilizers, please provide the formulas and %N and %P. For potassium fertilizers, please add %K.
Reply: Thank you for your suggestion. We have updated the description in Subsection 2.2 to include the chemical formulas and the percentage of active nutrients for nitrogen, phosphorus, and potassium fertilizers.
Subsection 2.3. Lines 130-131 Is this really a method for determining SOM? Or maybe this is how organic carbon is determined. The principle of determining available phosphorus and potassium should be added.
Reply: Thank you for your suggestion. I have revised the text in Subsection 2.3 to provide a more accurate and detailed description of these methods. Specifically, I have clarified that the method described is for determining soil organic carbon (SOC), which is then used to calculate SOM. Additionally, I have added the principles behind the determination of available phosphorus and potassium.
- Results -
Line 160 should be: Table 2. Cotton yields should be given in t ha-1 or Mg ha-1. Letter designations should be placed next to numerical data concerning yields. Please check the entire manuscript for superscripts and subscripts for the following forms: NH4+-N, NO3--N, and N2
Reply: Thank you for your detailed feedback on the presentation of the results. We have made the following revisions to address your comments.I have updated Table 2 to present cotton yields in t ha−1 (tonnes per hectare) and included letter designations next to the numerical data to indicate significant differences between treatments.I have reviewed the entire manuscript to ensure consistent formatting of chemical species, including NH4+-N, NO3--N, and N2.
- Discussion
The contents of subsection 4.3. should be saved below line 359.
Reply: Thank you for your suggestion. The content of Section 4.3 is to supplement and explain what deficiencies our research has and what aspects we should improve and study in the next step. It is more appropriate to put it in the discussion.
Specific remarks
- Line 55 should be: ...Liu et al. [13].....
- Line 129 and 136 …America.. Which America?
- Line 156 should be: …(Table 2)….
- Line 306 Should be: …Ma et al. [25]…
Reply: Thank you for your detailed comments and suggestions regarding the manuscript. I have made the following revisions to address each of your points:
- Line 55 Correction: We have corrected the citation format to read “Liu et al. [13]” to ensure proper referencing in the text.
- Lines 129 and 136 Clarification: Ihave clarified the reference to "America" by specifying it as "USA" in both instances. The revised text now reads: “...using an online monitoring system for three soil parameters (Campbell CS655, USA).”
- Line 156 Correction: Ihave updated the reference to the table to correctly state: “...with the overall performance as follows: Q90 > Q80 > Q100 (Table 2).”
- Line 306 Correction: Ihave corrected the citation to read “Ma et al. [25]” to ensure consistency and accuracy in our references.
5.The References section should be adapted to editorial requirements.
Reply: Thank you for your suggestion. I have reviewed and revised the References section to ensure that all citations are formatted according to the APA style guidelines, which are the editorial requirements for this journal. This includes ensuring proper punctuation, capitalization, and the inclusion of all necessary publication details for each reference. Additionally, I have checked the accuracy of the references and ensured that all cited works are correctly listed and formatted.
